# Long-Term Outcomes after Stroke in Patients with Atrial Fibrillation: A Single Center Study

**DOI:** 10.3390/ijerph20043491

**Published:** 2023-02-16

**Authors:** Justyna Tracz, Iwona Gorczyca-Głowacka, Anita Rosołowska, Beata Wożakowska-Kapłon

**Affiliations:** 1Clinic of Neurology, Swietokrzyskie Neurology Center, 25-736 Kielce, Poland; 21st Clinic of Cardiology and Electrotherapy, Swietokrzyskie Cardiology Centre, 25-736 Kielce, Poland; 3Collegium Medicum, Jan Kochanowski University, 25-317 Kielce, Poland

**Keywords:** atrial fibrillation, ischemic stroke, mortality, outcomes, stroke recurrence

## Abstract

Atrial fibrillation (AF) is known to be a significant risk factor for poor prognosis after stroke. In this study, we compared differences in long-term outcomes after ischemic stroke among patients with AF and sinus rhythm (SR). We identified patients admitted to the reference Neurology Center between 1 January 2013 and 30 April 2015, inclusive, with acute ischemic stroke. Of the 1959 surviving patients, 892 were enrolled and followed for five years or until death. We analyzed the risk of stroke recurrence and death between patients with AF and SR at 1, 3, and 5 years after stroke. The rates of death and stroke recurrence were estimated using Kaplan–Meier analysis and multivariate Cox regression. During follow-up, 17.8% of patients died and 14.6% had recurrent stroke. The mortality in the AF group increased relative to the SR group with subsequent years. The risk of death was statistically higher in the AF than SR group at 1 year after stroke (13.5 vs. 7%, *p* = 0.004). After adjusting for age, stroke severity, and comorbidities, there was also no significant effect of AF on mortality in the first year after stroke (OR = 1.59, *p* = 0.247). There were no significant differences between the groups in stroke recurrence during follow-up. The results of our study showed that post-stroke patients with AF have a more severe prognosis, although AF itself does not have an independent negative effect on long-term outcomes after stroke. Long-term survival after stroke in patients with AF was strongly associated with age, stroke severity, and heart failure. The impact of other factors on prognosis after stroke in patients with AF should be considered.

## 1. Introduction

Stroke is one of the most common causes of death in the adult population and the leading cause of disability worldwide. Recurrent stroke is becoming a significant medical and social problem due to increased risk of death and greater long-term disability [1,2]. Both stroke recurrence and long-term mortality also depend on other factors that increase stroke risk, such as age and comorbidities, and the applied secondary prevention [2,3].

Atrial fibrillation (AF) is the most common cardiac arrhythmia and is a major source of cardiogenic embolism, associated with an increased risk of stroke and systemic embolic events [4,5]. The presence of AF is associated with a five-fold higher incidence of stroke [5,6]. Patients with AF have worse neurological outcomes than patients with sinus rhythm (SR), and have a high recurrence rate and higher mortality. The one-year risk of death after stroke in patients with AF is twice that of those without AF [7]. However, reports on the long-term outcomes after stroke in patients with AF are scarce.

In this study, we compared long-term mortality and recurrent stroke in patients after ischemic stroke between patients with AF and those with SR. Such information may be useful to assess risk factors for death and recurrent stroke and the impact of secondary prevention in patients with AF.

## 2. Materials and Methods

### 2.1. Study Group

This study was a retrospective, observational, single-center study that included 2339 consecutively hospitalized patients in the reference Neurology Center between 1 January 2013 and 30 April 2015, inclusive, with a diagnosis of acute ischemic stroke.

Ischemic stroke was defined according to the World Health Organization (WHO) and brain imaging by computed tomography (CT) or magnetic resonance imaging (MRI) [8]. All types of ischemic stroke were included in the study. Patients with cerebral hemorrhage and transient ischemic attack (TIA) were excluded from this study.

Patients were categorized into 2 groups according to the presence of AF (AF group) or SR (SR group).

AF was defined as a history of AF or AF diagnosed during the hospitalization by electrocardiography (ECG) or 24 h monitoring electrocardiography (Holter ECG).

We compared patient characteristics in the AF and SR group and followed up prognosis at 1, 3, and 5 years after stroke.

Clinical data were collected by interview and analysis of medical records.

Demographic data (sex, age), comorbidities’ diagnostic study results, and hospital course were analyzed. Risk factors for stroke included hypertension, diabetes mellitus, heart failure, coronary artery disease, previous stroke or TIA, internal carotid artery (ICA) stenosis >70% (assessed with a Doppler ultrasound), alcohol consumption, and current smoking. Antithrombotic treatment was assessed via follow-up interviews.

Stroke severity was assessed using the National Institutes of Health Stroke Scale (NIHSS) at admission and at discharge, and was categorized into three groups: mild (NIHSS score <8), moderate (NIHSS score 8–16), and severe (NIHSS score >16) [5,9].

To assess thromboembolic risk, CHA2DS2-VASc scores were calculated for all patients after stroke, according to the European Society of Cardiology (ESC) 2020 guidelines on AF. Low thromboembolic risk patients were classified as having a score of 0 (1 in women), intermediate thromboembolic risk patients as having a score of 1 (2 in women), and high thromboembolic risk patients as having score ≥2 (≥3 in women) [10].

The protocol of this study was consistent with the Declaration of Helsinki and was approved by the Ethics Committee of Swietokrzyska Medical Chamber.

### 2.2. Follow up and Outcomes

Of all 2339 patients, 380 (16.3%) died during the hospital stay. Among those patients who died, 115 had AF (30.3%). The remaining 1959 survivors were further analyzed. These patients were followed up for up to 5 years after the index stroke or until death. Data were assessed by a neurologist by phone. If information could not be obtained from the patient, the interview was collected from the legal representative or family. Follow-up interviews included questions about recurrent vascular events, disability, and treatment.

Clinical outcomes of this study were incidence rates of mortality and stroke recurrence at 1, 3, and 5 years after stroke for patients with AF compared to patients with sinus rhythm.

Morality was defined as all-cause cumulative death over a given time period during the follow-up.

Recurrent stroke was defined as a new neurological deficit lasting >24 h, occurring after a baseline stroke and confirmation of new ischemic changes on CT or MRI [11,12].

### 2.3. Statistical Analysis

In order to compare the AF and SR groups in terms of ratios of the variables tested, a series of comparison analyses were performed using the chi-square test and Student’s t test for independent samples. The chi-square test of independence was performed for qualitative and categorical variables, which were compared for differences in the proportions of outcomes occurring within the two study groups. For quantitative variables, comparisons were made based on the mean level of variables in each group with the assumption of a 95% confidence interval of the true result for the Student’s t test. A significance threshold of α = 0.05 was assumed for each test. Analyses were performed using IBM SPSS Statistics 27, in which relationships between AF and mortality and risk of recurrent stroke were tested by logistic regression analysis, and results were presented as unadjusted odds ratios (ORs) with 95% confidence intervals (CIs). Multivariate logistic regression analysis was performed with categorical and qualitative factors that were significantly different between groups. A threshold value of α = 0.05 was considered statistically significant. Stroke mortality and recurrence in the AF and SR groups were also analyzed using Kaplan–Maier survival curves along with the Cox hazard ratio (HR) to calculate the odds ratio of stroke survival and recurrence with subsequent years and to confirm the differences in these two variables according to the presence of AF. A threshold of α = 0.05 was used as the level of significance. To assess the factors determining mortality among SR and AF, logistic regression analyses were performed. These results were presented using unadjusted odds ratios (ORs) with 95% confidence intervals (CIs). A threshold of α = 0.10 was also adopted to indicate potential risk factors.

## 3. Results

### 3.1. Baseline Characteristic

Of the 1959 patients who survived the initial stroke, 1067 patients were excluded whose health information could not be obtained due to missing or incomplete observational data, or if they had not consented to participate in the study. In total, 892 patients with stroke were enrolled in the study, including 223 (25.1%) with AF and 669 (74.9%) with SR (Figure 1).

The mean age was 70 ± 12.22 years, and 489 patients (54.8%) were male. Compared to patients in the SR group, patients with AF were older (75.37 ± 9.46 vs. 69.17 ± 12.64), were more often female (59% vs. 41%, *p* < 0.001), and had higher frequency of heart failure (23 vs. 7%, *p* < 0.001), ischemic heart disease (42 vs. 27%, *p* < 0.001), hypertension (84 vs. 76%, *p* = 0.013), and history of stroke or TIA (20 vs. 14%, *p* = 0. 018). There were no significant differences in the prevalence of diabetes, hyperlipidemia, peripheral atherosclerosis, and significant ICA stenosis between these two groups. However, patients in the SR group were significantly more likely to smoke cigarettes (18 vs. 5%, *p* < 0.001) and consume alcohol (11 vs. 4%, *p* = 0.002) than in the AF group.

From the data collected during follow up, oral anticoagulation (OAC) was used by 38% of participants in the AF group.

The NIHSS score at admission and at discharge was significantly higher in the AF group compared to the SR group (mean, 7.03 ± 5.38 vs. 5.27 ± 5.08; 4.16 ± 4.63 vs. 3.09 ± 3.79 *p* < 0.001), and more severe strokes (NIHSS >16) were observed in the AF group (13 vs. 7% at admission and 5 vs. 1% at discharge, *p* < 0.001).

The risk of stroke in patients with AF expressed by the CHA2DS2-VASc scale was high (mean, 5.31 ± 1.50).

Clinical characteristics of study groups are shown in Table 1.

### 3.2. Outcomes

Of all 892 patients, 159 (17.8%) died during the 5-year follow-up period, and 130 (14.57%) suffered a recurrent stroke.

The cumulative incidence of death in the AF group compared to the SR group was 13.5 vs. 7%, 21.1 vs. 11.9%, and 23.1 vs. 15.9% at 1, 3, and 5 years, respectively, and the cumulative incidence of recurrent stroke was 10.3% vs. 7.2%, 14.3 vs. 10.9%, and 17.0 vs. 13.7% at 1, 3, and 5 years, respectively. Kaplan–Meier survival curves were used to evaluate differences in the incidence of death and recurrent stroke in subsequent years among patients with AF and SR. The analysis showed that the risk of death increased in subsequent years in the AF group while there were no significant differences in the risk of recurrent stroke in subsequent years (Figure 2).

In univariate analysis, 1-year mortality was significant higher in the AF group compared to the SR group (13.5 vs. 7%, *p* = 0.004). There were no significant differences between the AF group and the SR group in morality at 3 years (7.6 vs. 4.9%, *p* = 0.133) and 5 years after stroke (2.2 vs. 4%, *p* = 0.219). There were no differences in the frequency of stroke recurrence at 1 year (10.3 vs. 7.2, *p* = 0.136), 3 years (4 vs. 3.7%, *p* = 0.840), and 5 years after stroke (2.7 vs. 2.8, *p* = 0.907). Multivariate analysis including gender, age, medical history, and stroke severity showed no significant effect of AF on mortality at 1 year after stroke (Table 2).

## 4. Discussion

In this study, we assessed long-term outcomes after ischemic stroke among patients with AF compared with those in the SR group. The main observations of our study are as follows. The group with AF had a more severe prognosis. We found a higher mortality rate after ischemic stroke among patients with AF, with a significant difference occurring in the first year after stroke. AF itself was not observed to be a predictor of death and stroke recurrence in post-stroke patients during long-term follow-up. The worse prognosis in the AF group was mainly related to older age and multimorbidity.

The findings from previous studies have shown that AF is associated with a worse prognosis in patients with ischemic stroke, but the research mainly assessed short-term mortality and reports of stroke recurrence in patients with coexisting AF are rare [13,14,15,16].

In our study, patients in the AF group were older than those in the SR group. Among patients with AF, those over 74 years of age accounted for the largest proportion, which is similar to the results of other studies [16,17,18,19].

Compared to patients with SR, we observed a higher frequency of hypertension, heart failure, coronary heart disease, and history of cerebral incident in the AF group, but a lower prevalence of smoking and alcohol consumption. The most common risk factors for stroke in patients with AF are those included in the CHA2DS2-VASC score (heart failure, hypertension, diabetes, previous stroke, vascular disease, gender) and others such as hyperlipidemia or smoking [20,21]. Reports on risk factors for stroke in patients with AF are inconsistent. The Lamassa et al. study showed that patients with AF were more likely to have a history of myocardial infarction, but less likely to have risk factors for stroke such as diabetes, smoking, and alcoholism [15]. Similarly, Marini et al. showed that patients with AF were more likely to have coronary artery disease and peripheral artery disease, but were less likely to be smokers [17]. In another study, there was no difference between the prevalence of comorbidities in the two groups, except for the higher prevalence of diabetes in the SR group [5].

Our study showed that patients with AF had more severe strokes, which is consistent with other reports [5,14,16]. Regarding NIHSS scores at admission and at discharge, statistically significant differences were found, which indicated a higher percentage of NIHSS <7 in the SR group and NIHSS >16 in the AF group. Stroke severity is considered an important factor primarily for early death, but it also affects long-term prognosis after stroke [22,23].

According to some authors, 1-year survivors die within the next 4 years at a rate of approximately 10% per year, with recurrence rates of approximately 10% within the first year after stroke and 5% per year thereafter [5,24]. In other reports, the risk of recurrent stroke in patients with AF is 10% per year, with mortality rates as high as 50% within 2 years after stroke [25,26].

The results of our study on the effect of AF on long-term prognosis after stroke showed that the difference in mortality in the AF group increased relative to the SR group with subsequent years. Analyzing successive follow-up periods (1, 1–3, and 3–5 years after stroke), we found that a significant difference appeared in the first year, and then mortality was not significant in subsequent time periods. There was no significant effect of AF on poorer outcomes in stroke patients after adjusting for other factors. There was also no difference in the rate of stroke recurrence in the AF and SR groups during follow-up. Similar results were obtained by Zhao et al., who reported that mortality and stroke recurrence rates were not different at 1 and 3 years [5]. Wang et al. showed that after 1 and 3 years the risk of death was higher in older patients with AF, but not the risk of recurrent stroke [9]. Another study observed higher recurrence and mortality in the AF group at 1 year after stroke [14], which is similar to the study by Yang et al., who showed that patients with both previously diagnosed and newly diagnosed AF were at higher 1-year risk of stroke recurrence and mortality [27].

The characteristics of the study population, the prevalence of risk factors and comorbidities, and differences in the definition of recurrent stroke may influence the results of individual studies. Other possible factors contributing to poor prognosis in stroke patients should be investigated. Survival and incidence of recurrence depend largely on functional capacity after stroke, comorbidities, and the effectiveness of secondary prevention.

The risk of death is highest in the first post-stroke weeks and is usually related to the stroke itself, but this risk persists long after stroke, due to recurrence and the presence of other risk factors, mainly cardiovascular disorders [13,23,24,25].

Comparing the determinants of mortality in the SR and AF groups, we found no significant risk factors for increased mortality in patients with AF in the first year after stroke, suggesting that AF may have been the primary cause. In subsequent years, long-term poststroke survival in patients with AF was strongly associated with advanced age, stroke severity (NIHSS >16), and heart failure at 5 years after stroke (Table 3).

These results are not surprising, as older patients and those with more severe strokes are more prone to complications and poorer outcomes. Age is an important factor that increases the risk of both AF and ischemic stroke. Some reports suggest that older stroke patients have increased short- and long-term mortality, a higher risk of stroke recurrence, and more frequent disability [9,17]. In addition, older patients may be at greater risk for poorer outcomes due to a higher prevalence of comorbidities. In our analysis, heart failure was an additional factor contributing to higher mortality at 5 years after stroke in patients with AF. It is a common risk factor for both stroke and AF. The presence of heart failure is associated with a poor prognosis in stroke survivors, and post-stroke mortality increases significantly with coexisting AF [28]. Stroke severity is considered the most important factor influencing outcome after stroke [22,23]. In our analysis, stroke severity at discharge was most strongly associated with risk of death between 3 and 5 years after stroke, indicating that its impact extends even beyond the initial stroke period. We were surprised to find that the prognosis after stroke appeared to be better in patients with greater neurological deficits on admission than in patients with minor deficits. At the discharge, the relationship was reversed, with mortality increasing with NIHSS >16. We think that these differences can be explained, in part, by the fact that better hospital care in patients with more severe stroke on admission reduced the risk of further complications, whereas, after discharge, it is likely that lack of medical care was a major determinant of mortality. At 5 years, stroke severity at discharge had less impact on death, likely related to the fact that most patients with severe stroke died within the first 3 years after the stroke. In addition, other factors, including age and heart failure, influenced the prediction of death.

It is important to note that, during the study period, significant advances have been made in the treatment of stroke through the increased use of thrombolytic therapy and the evolution of intravascular mechanical thrombectomy in the acute phase of stroke, improved access to post-stroke rehabilitation, and the increasing use of NOAC in secondary prevention in patients with AF. Therefore, some studies have reported increasingly better survival rates for patients with AF after stroke [26]. In addition, there has been a significant increase in diagnostics for previously undetected AF after stroke, such as long-term ECG monitoring with mobile ambulatory cardiac telemetry or implantable devices. There are also reports that have shown that a new ECG parameter, P-wave peak time, may be useful in predicting the risk of developing paroxysmal AF among patients with acute ischemic stroke [29,30]. Early implementation of effective cardiovascular incident prevention strategies may improve long-term survival after stroke.

### Limitations

There are several limitations of this study. First, it was a retrospective observational study conducted at a single hospital center, so the AF patient population was limited. In addition, a significant limitation was that the proportion of patients who were lost to follow-up was high (50%) and could limit the results of the study. Differences in the definition of stroke recurrence may have influenced the study results. Another limitation was that asymptomatic paroxysmal AF could go unrecognized during the study period, increasing the risk of unrecorded AF. There were no data on the type of stroke (embolic or atherosclerotic). Data on anticoagulant treatment were incomplete, resulting from its retrospective character, and the percentage of AF patients treated with anticoagulants was much lower than today, with widespread use of NOAC. Furthermore, it is unclear why higher NIHSS scores in the AF group had a different impact on mortality at admission and at discharge 3 years after stroke. It is possible that this discrepancy between the results was influenced by other factors, such as in-hospital treatment during the acute phase of the stroke, rehabilitation, and follow-up medical visits after discharge. We did not have information on medical care among our patients or data on the cause of death after discharge, which limits the interpretation of factors affecting mortality. Due to the single-center design and small sample size of our study, these results should be treated with caution and prospective studies with larger samples are needed to confirm our findings.

## 5. Conclusions

Patients with AF had a more severe post-stroke prognosis, with a higher mortality rate after ischemic stroke than those with SR. However, AF itself had no independent effect on mortality or stroke recurrence. Among the factors associated with long-term survival after ischemic stroke in AF patients, older age, stroke severity, and heart failure were the strongest. The impact of other factors on post-stroke prognosis in patients with AF should be considered, and the importance of effective treatment of identified modifiable risk factors in addition to antithrombotic therapy should be emphasized.

## Figures and Tables

**Figure 1 ijerph-20-03491-f001:**
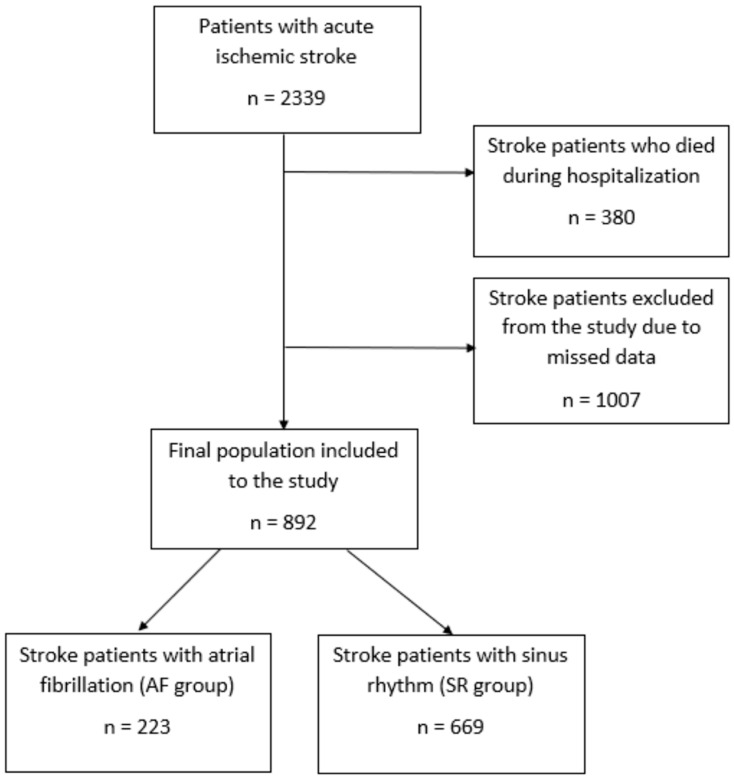
The flow chart of the study.

**Figure 2 ijerph-20-03491-f002:**
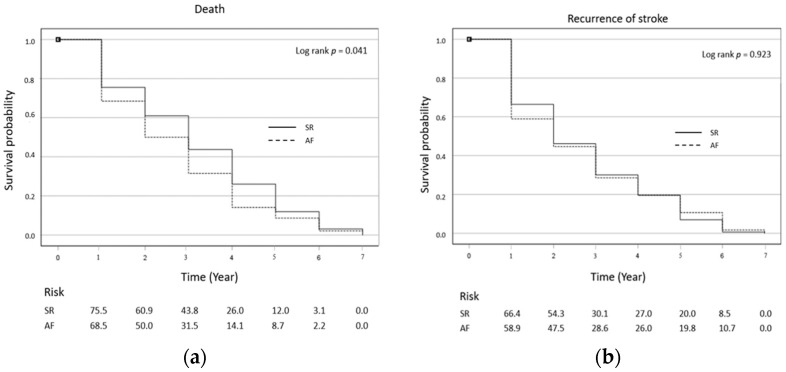
Kaplan–Meier curves of stroke morality (**a)** and stroke recurrence (**b**) in the AF and SR groups.

**Table 1 ijerph-20-03491-t001:** Clinical characteristics of the study group.

Characteristic	All Patients *n* = 892	AF Group *n* = 223	SR Group *n* = 669	*p* (x²)
Age (years)	70.7 (12.22)	75.4 (9.46)	69.2 (12.64)	0.001
Age group, *n* (%)				<0.001
<65 years	278 (31)	30 (13)	248 (37)	
65–74 years	225 (25)	64 (29)	161 (24)	
>74 years	389 (44)	129 (58)	260 (39)	
Gender, *n* (%)				
Females	403 (45)	131 (59)	272 (41)	
Males	489 (55)	92 (41)	397 (59)	
Medical history, *n* (%)				
Heart failure	96 (11)	51 (23)	45 (7)	<0.001
Hypertension	699 (78)	188 (84)	511 (76)	0.013
Diabetes mellitus	218 (24)	58 (26)	160 (24)	0.529
Previous stroke/TIA	136 (15)	45 (20)	91 (14)	0.018
Coronary artery disease	275 (31)	93 (42)	182 (27)	<0.001
Dyslipidemia	151 (17)	37 (17)	114 (7)	0.877
Peripheral arterial disease	42 (5)	13 (6)	29 (4)	0.364
Smoking	130 (15)	12 (5)	118 (18)	<0.001
Alcoholism	83 (9)	9 (4)	74 (11)	0.002
ICA stenosis >70%	129 (15)	24 (11)	105 (16)	0.062
Laboratory tests, *n* (%)				
HGB < 12 g/dl	84 (9)	32 (14)	52 (8)	0.004
PLT < 150 g/dl	70 (8)	23 (10)	47 (7)	0.114
Thrombolytic therapy	66 (7)	17 (8)	49 (7)	0.883
Antithrombotic treatment *, *n* (%)				<0.001
Antiplatelet agents	506 (78)	85 (56)	421 (85)	
VKA	24 (4)	15 (1)	9 (2)	
NOAC	82 (13)	43 (28)	39 (8)	
no medication	38 (6)	9 (6)	29 (6)	0.848
NIHSS score (mean ± SD)				
at admission	3.35 (±4.04)	7.03 (±5.83)	5.27 (±5.08)	<0.001
at discharge	3.36 (±4.04)	4.16 (±4.63)	3.09 (±3.79)	<0.001
NIHSS at admission				<0.001
≤7	601 (69)	130 (60)	471 (72)	
8–15	196 (23)	59 (27)	137 (21)	
≥16	70 (8)	27 (13)	43 (7)	
NIHSS at discharge				<0.001
≤7	702 (82)	161 (76)	541 (85)	
8–15	134 (16)	41 (19)	93 (15)	
≥16	16 (2)	10 (5)	6 (1)	
CHA_2_DS_2_-VASc (mean ± SD)	4.65 ± 1.57	5.31 ± 1.50	4.43 ± 1.54	<0.001

* Data available for 650 patients (152 in the AF group and 498 in the SR group). Abbreviations: AF, atrial fibrillation; HGB, hemoglobin; ICA, internal carotid artery; PLT, platelets; NIHSS, National Institutes of Health Stroke Scale; NOAC, novel oral anticoagulants; SR, sinus rhythm; TIA, transient ischemic attack; VKA, vitamin K antagonists.

**Table 2 ijerph-20-03491-t002:** Univariate and multivariate logistic regression analysis outcomes at 1 year, 3 years, and 5 years after stroke among patients with AF and SR.

Outcomes	AF Group	SR Group	Univariate Analysis		Multivariate Analysis	
			OR (95% CI)	*p*	OR (95% CI)	*p*
1 Year						
Mortality	30 (13.5)	47 (7.0)	2.06 (1.27–3.34)	0.004.	1.59 (0.72–3.51)	0.247
Recurrence	23 (10.3)	48 (7.2)	1.49 (0.88–2.51)	0.136	1.85 (0.79–4.33)	0.157
3 Years						
Mortality	17 (7.6)	33 (4.9)	1.59 (0.87–2.92)	0.133	0.82 (0.31–2.15)	0.683
Recurrence	9 (4.0)	25 (3.7)	1.08 (0.50–2.36)	0.840	1.36 (0.35–5.34)	0.660
5 Years						
Mortality	5 (2.2)	27 (4.0)	0.55 (0.21–1.43)	0.219	0.36 (0.07–1.76)	0.207
Recurrence	6 (2.7)	19 (2.8)	0.95 (0.37–2.40)	0.907	0.19 (0.03–1.18)	0.075

Abbreviations: AF, atrial fibrillation; OR, odds ratio; SR, sinus rhythm. Multiple regression with all variables is shown in Appendix A.

**Table 3 ijerph-20-03491-t003:** Factors determining mortality among SR and AF.

Factor	SR Group		AF Group	
	OR (95%CI)	*p*	OR (95%CI)	*p*
1 Year				
>75 years	3.33 (1.37;8.14)		0.008	—
Hypertension	0.51 (0.25–1.07)		0.074	—
3 Years				
>75 years	3.02 (1.52–6.00)	0.002	10.84 (1.27–92.41)	0.029
NIHSS at admission > 16	—		0.16 (0.02–1.20)	0.075
NIHSS at discharge > 16	—		8.37 (0.80–87.66)	0.076
5 Years				
66–74 years	2.74 (1.36–5.53)	0.005	—	
>75 years	4.47 (2.34–8.53)	<0.001	4.69 (0.87–25.26)	0.072
Heart failure	—		2.07 (0.91–4.73)	0.083
NIHSS at admission > 16	—		0.18 (0.03–1.27)	0.085

Abbreviations: AF, atrial fibrillation; OR, odds ratio; SR, sinus rhythm; NIHSS, National Institutes of Health Stroke Scale.

## Data Availability

Not applicable.

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
