# Peer review of "Long-Term Outcomes after Stroke in Patients with Atrial Fibrillation: A Single Center Study"

_ijerph, 2023, doi:10.3390/ijerph20043491_

Round 1

Reviewer 1 Report (Previous Reviewer 1)

I read with interest the next version of the article by Tracz et al. "Long-term outcomes after stroke in patients with atrial fibrillation. A single center study". This time I tried to abstract from my acquaintance with previous versions and my previous comments. Nevertheless, even with this approach, I had questions for the authors of the article, to which I would like to receive answers.

1. The authors in the Conclusion indicate "Patients with AF had a higher mortality rate after ischemic stroke than those without AF, although the significant difference appeared only in the first year after stroke." However, based on Table 1, it can be seen that patients with AF were older, more often had concomitant CHF and IHD, they had a higher severity of stroke both at admission and at discharge. That is, in the group of patients with AF, there were many factors that aggravated the prognosis after a stroke. Not surprisingly, AF had an impact on mortality 1 year after stroke only in a univariate logistic regression analysis. And when considering other factors, the authors rightly write: "Multivariate analysis including gender, age, medical history, stroke severity showed no significant effect of AF on mortality at 1 year after stroke". Accordingly, the first sentence in the Conclusion is incorrect. The point is not the presence or absence of AF in patients, but the fact that the group with AF is more severe in terms of prognosis, and AF itself does not have an additional negative impact on the prognosis. This is how the emphasis should be placed in the Conclusion.

2. The authors provided in the manuscript Table S1. "Multiple regression of outcomes at 1 year, 3 years and 5 years after stroke with all variables". These are interesting data, however, to answer the question whether AF has an independent effect on long-term outcomes after stroke, the presence of AF in patients should be included in the multiple logistic regression model.

3. The association of stroke severity with prognosis in patients in the AF group remained a mystery to me (Table 3). Why did the presence of a severe stroke at admission (NIHSS at admission >16) have a clear association with an improvement in prognosis after 3 years (OR 0.16), although not reaching statistical significance (p = 0.075), and the severity of stroke at discharge (NIHSS at discharge > 16), on the contrary, significantly worsened the prognosis after 3 years (OR 8.37), although also with marginal significance (p=0.076). I would like to understand what happened to patients during treatment in a hospital, what so radically changed the prognosis?

4. Section Discussion should not start with literary references, but with the main result obtained in the study. Therefore, the first paragraph of this section should be moved either to the Introduction section or to the subsequent text of the Discussion section.

Round 2

Reviewer 1 Report (Previous Reviewer 1)

The authors really did a great job of correcting the manuscript. However, their response to my remark 3 still does not satisfy me. In my opinion, it cannot be that the severity of the stroke at the time of admission and at the time of discharge affects the prognosis in the opposite way. Either there are errors in the statistical processing of the results, or in design features (evaluation of stroke outcomes separately at different time intervals, and not on an accrual basis), or the version of the authors is really correct. In any case, the authors' reasoning in response to remark 3 should be reflected in the Study Limitations section.

Author Response

This manuscript is a resubmission of an earlier submission. The following is a list of the peer review reports and author responses from that submission.

Round 1

Reviewer 1 Report

I would like to thank the authors for the work done to correct the manuscript, which allowed them to better present the results. Nevertheless, I have to admit that even in the revised version of the manuscript there are significant problems that need to be eliminated.

1.       On my first question. I would like to thank the authors for providing data on the severity of stroke in the AF and SR groups according to the NIHSS scale at admission and at discharge from the hospital. These data confirmed my assumption that the severity of stroke was higher in the AF group (I would like to note that I did not claim that all patients with AF had a cardioembolic stroke; they have it more often than in the SR group). Based on these data, it may well turn out that the prognosis in patients after a stroke is affected not by AF, but by the severity of the stroke associated with this arrhythmia. This assumption can be confirmed or refuted by conducting multiple regression analysis with the inclusion of stroke severity according to the NIHSS scale as covariates. But the authors did not do this, they took into account other variables (gender, age, medical history).

2.       On my second question. The authors provided a cumulative analysis of events (death and recurrent stroke) over a 5-year follow-up period. This Kaplan-Meier analysis showed that in the AF group, survival was worse, and the groups did not differ in the frequency of recurrent strokes (judging by the initial differences in the severity of strokes in the groups, the result is expected). However, the authors still retained the results of the regression analysis taking into account individual events in the periods 1-3 years and 3-5 years after stroke. In my opinion, there is no clinical sense in the analysis of such time intervals - what difference does it make that in the AF group in the period 3-5 after a stroke, patients die no more often if they died at an earlier date and fewer of them lived to this period of time. Other researchers usually study the cumulative event rate in prospective observation. For example, I can recommend authors to look at a recent article by Yang et al. (1). I am sure that with this version of the regression analysis, the authors can get completely different conclusions.

3.       Regarding my third question. In my opinion, the retrospective nature of the study cannot influence the diagnosis of the type of stroke in patients. In such studies, it is impossible to influence the treatment and diagnostic tactics, but it is quite possible to obtain data from the medical documentation about the diagnosis made by the patient. Or in this neurological center they do not adhere to the practice of specifying the type of stroke without fail? I think that the lack of information about the type of stroke in the examined patients should be included in the Study Limitations section.

4.       The answers of the authors to my questions 4-5 satisfied me, there are no additional comments.

5.       However, a question arose about the new version of Table 3. When analyzing the factors determining mortality among FA sroup in the period 1-3 years after a stroke, different effects on the prognosis of the NIHSS scores attract attention: when it enters, its values of more than 16 points in admission improve the prognosis (OR 0.16 ), and worsened at discharge (OR 8.37). How can the authors explain this? The same question for the period from 3 to 5 years - does the prognosis improve with severe stroke (OR 0.18)? I only ask the authors not to answer that the data were not statistically significant, the trends are quite clear!

References:

1.     Yang F, Yan S, Wang W, Li X, Chou F, Liu Y, Zhang S, Zhang Y, Liu H, Yang X, Gu P. Recurrence prediction of Essen Stroke Risk and Stroke Prognostic Instrument-II scores in ischemic stroke: A study of 5-year follow-up. J Clin Neurosci. 2022 Oct;104:56-61. doi: 10.1016/j.jocn.2022.07.011.

Reviewer 2 Report

I feel the manuscript adds to the existing literature in stroke and atrial fibrillation. 

Author Response

Thank you for your opinion.

Round 2

Reviewer 1 Report

At this stage, the authors practically did not make changes to the text of the manuscript, limiting themselves only to answers on my comments. Unfortunately, these answers showed that the authors did not quite understand the presented statistical data.

On the first question, it is desirable to present in Table 2 data on all indicators included in the multiple logistic regression model (gender, age, medical history, stroke severity). By the way, it is not clear why the inclusion of an additional parameter in the model (stroke severity) did not affect the data presented in Table 2 in any way compared to the previous version.

On the fifth question. The authors incorrectly interpret the OR indicator. The values of this indicator of 0.16 do not indicate a "16% higher risk of death", but that the risk of death is only 16% of the risk of death in the group with NHISS values > 16. To be honest, it is difficult for me to imagine this clinical situation. An analogous situation with OR values of 0.18 in the analysis of mortality after 5 years.
